# *Prevotella copri*-mediated caffeine metabolism involves ferroptosis of osteoblasts in osteoarthritis

Feng Li,[1] Xin Wen,[1] Pu Xue,[1] Huiping Xu,[1] Panyang Wu,[1] Zhiming Xu,[2] Xianwei Wang,[1] Guofu Pi[1]

**ABSTRACT**    There is a positive causality between coffee consumption and osteoarthritis (OA); however, whether gut microbiota is involved needs to be discussed. Here, we observed that in caffeine consumers, fecal *Prevotella copri* abundance was positively correlated with subchondral bone mass, serum caffeine concentration was negatively correlated with bone mass, and fecal *P. copri* was negatively correlated with serum caffeine. In the OA model, caffeine intake aggravated articular cartilage destruction, bone mass loss, and intestinal barrier damage; on the contrary, paraxanthine intake reversed the above lesions. Importantly, after the intestinal *P. copri* supplement, caffeine-induced lesions in OA mice were effectively alleviated. Mechanically, *P. copri* has the potential to metabolize caffeine into paraxanthine, and this effect could alleviate the ferroptosis of osteoblast in the OA model. This study screened out that *P. copri,* an endogenous bacteria, has the ability to metabolize caffeine and revealed its effects on OA progression.

**IMPORTANCE**    There is positive causality between coffee consumption and osteoarthritis (OA). Caffeine exposure is responsible for the reduction of bone mass and restrained osteoblast function. *Prevotella copri* abundance is exhausted in gut and positively correlated with subchondral bone mass in coffee consumption patients with OA. Supplement of intestinal *P. copri* alleviates caffeine-induced subchondral bone loss. *P. copri* has the potential to metabolize caffeine into paraxanthine, and this effect alleviates ferroptosis of osteoblast. Our study illustrated that intestinal *P. copri* possibly serves as a novel promising treatment for coffee consumers with OA.

**KEYWORDS**    osteoarthritis, *Prevotella copri*, caffeine, subchondral bone, ferroptosis

C offee is a regular part of the diet of many adults and is consumed by a certain number of the world's population, especially Western people (1). Caffeine, the key component of coffee, is the world's most widely used drug and one of many constituents in food, which can exert physiologic effects, some of which may be pathological (2). Subchondral bone loss is a vital precipitating factor for the destruction of articular cartilage in osteoarthritis (OA) progression (3). Bone homeostasis refers to the stable state of bone metabolism, that is, osteoclast-mediated bone resorption and osteoblast-mediated bone formation maintain the dynamic balance of bone shape and structure in the same bone reconstruction unit (4, 5). Tibial subchondral bone loss could be used as a predictor for the progression of knee OA (6, 7). Indeed, previous studies showed a positive causality between caffeine intake and OA progression (8–10). Caffeine exposure-induced susceptibility to OA was likely related to the reduction of subchondral bone mass that was due to the developmental restrained function of osteoblast differentiation (11). Importantly, the metabolic rate of caffeine affects the bone mineral density (BMD) of the human body, and previous reports also have shown that gut microbiota is involved in caffeine metabolism (12–14). Recently, the correlation between gut microbiota and

**Peer Reviewer** Jiacan Su, Shanghai Changhai Hospital, Second Military Medical University, Shanghai, China

Address correspondence to Guofu Pi, guofupi@yeah.net, or Xianwei Wang, wangxw807@126.com.

Feng Li, Xin Wen, and Pu Xue contributed equally to this article. The author order was determined by the magnitude of the authors' contribution, with those who contributed the most coming first.

The authors declare no conflict of interest.

bone metabolism has gradually become a widespread research hotspot, and successive studies have revealed that the alterations of gut microbiota are related to the occurrence and progression of age-related degenerative diseases of bone (15–17). However, whether gut microbiota is involved in the positive causality between caffeine intake and OA progression has not been elucidated. Therefore, this study intends to explore the role of gut microbiota in caffeine metabolism and its effect on subchondral bone loss, in order to explain the mechanism of caffeine-related OA progression.

## MATERIALS AND METHODS

### Participants

We included a total of 79 subjects in the study, 39 subjects in the coffee group who drank coffee, about 300 mg per day, at least 5 days per week, and the non-coffee group of 40 subjects who never drank coffee. Prior to study inclusion, each participant provided written informed consent for research use and publication of their data.

### Cell source

MC3T3-E1 cell line was purchased from Shanghai Zhongqiao Xinzhou Life Technology Co., LTD. During the process of cell culture, mycoplasma detection was carried out regularly, and cell experiments were carried out after confirming that there was no mycoplasma infection.

### Animal

All animal experiments were ethically approved by the Laboratory Animal Center of the First Affiliated Hospital of Zhengzhou University. In addition, 8- to 10-week-old specific pathogen-free (SPF) C57BL/6 female mice, five in a cage with ventilation function of translucent plastic cages, strictly under 12 h of lighting and 12 h of darkness (lights on at 7 am every day). All mice were given a week to acclimate to their new environment before undergoing surgery.

### *Prevotella copri*

*Prevotella copri* strains were purchased from BeNa Culture Collection. Product No. BNCC354512 belongs to the lyophilized anaerobic strain.

### Destabilization of the medial meniscus (DMM)

The mouse OA model was caused by the destabilization of the medial meniscus (DMM) through surgery. In brief, the skin of the mice was sterilized with 10% povidone-iodine, and the mice underwent sterile surgery by intraperitoneal injection of 2% (wt/vol) pentobarbital (40 mg/kg) under anesthesia. The knee joint capsule was cut inside the patellar tendon, and the tibial ligament of the medial meniscus was cut with microsurgery scissors. After surgery, all mice were injected with penicillin 50,000 U/d for 3 consecutive days and recovered for 1 week.

### Gavage

Before gavage, a tube of frozen bacteria was removed from the refrigerator at $-80^{\circ}$C, melted at room temperature, and transferred to a sterile medium. After absolute anaerobic culture at 37°C for 48 h, the passaged bacteria were incubated at $37^{\circ}$C for 20–24 h until the concentration of bacteria solution reached $1 \times 10^{10}$ CFU/mL.

### Micro-CT

The knee of the mice was removed after death and placed in a 4% tissue-fixative solution for 48 h. After fixation, Skyscan 1176 Micro-CT instrument (Bruker micro-CT, Kontich,

Belgium) was used to scan the bone microstructure and bone mineral density. The parameters were set as follows: source voltage, 50 kV; source current, 500 µA; AI, 0.5 mm filter; pixel size, 9 µm; and rotation step, 0.4 degree. All images obtained after scanning were reconstructed using NReconru software (Bruker micro-CT, Kontich, Belgium) with the following reconstruction parameters: ring artifact correction, 7; smoothing, 2; and beam hardening correction, 40%. The analysis area was set 0.5 mm above the growth plate, and the analysis thickness was 1 mm. The region of interest (ROI) was determined by manual delineation, and the bone characteristics in the ROI were determined by the image binary method.

## Bone histological analysis

At the end of micro-CT analysis, the knee was immersed in 14% ethylene diamine tetraacetic acid (EDTA) solution and decalcified in a water bath at 37°C for 7 days, with the solution changed every 2 days. After decalcification, the sections were embedded in paraffin, and then for hematoxylin-eosin (HE) staining and immunofluorescence staining. The staining process was carried out strictly in accordance with the instructions. After the staining, Olympus BX51 microscope (Olympus Corporation, Takachiho, Japan) was used to observe and take photos. Bone histology images were analyzed using BIOQUANT OSTEO software (Bioquant Image Analysis Corporation, Nashville, TN, USA).

## Serum analysis

Caffeine and its metabolites were gavaged for 8 weeks, and *P. copri* was gavaged for 8 weeks. After gavage, blood samples were collected by cardiac puncture, and the serum was separated and then placed in a refrigerator at −80°C for subsequent detection.

## Quantitative PCR

The tissue was dissected and put into liquid nitrogen snap freeze. After packaging, the tissue was immediately stored in a refrigerator at −80°C for storage, and then, the tissue total RNA was extracted, reverse-transcribed, and qPCR was performed. The relative expression levels of target genes were measured by the ΔΔCT method.

## Fecal metagenome sequencing

Human fecal samples were flash-frozen in liquid nitrogen and then stored at −80°C in the refrigerator for later use. Fecal DNA was extracted using the QIAamp PowerFecal Pro DNA Kit following the manufacturer's instructions. Using a spectrophotometer with 1% agarose gel electrophoresis, the total DNA quality was measured. Using KAPA HyperPlus PCR-free, we broke up the DNA to an average size of 350 bp to construct a paired-end library. Metagenomic sequencing was then performed on an Illumina HiSeq platform according to the manufacturer's protocol. After we obtained all raw metagenomic sequencing data, we used MOCAT2 software to control the quality of these data. Using the SolexaQA package, raw sequence reads with lengths shorter than 30 bp, and quality scores lower than 20 were filtered out. Using SOAPaligner, we compared the filtered reads with the human genome to remove contaminated reads and obtain clean raw reads. Using SOAPdenovo software, we assembled the clean raw reads to obtain contigs for later annotation and prediction. We then used MetaGeneMark to predict the scaftigs of each sample.

## Serum metabonomics

Blood samples were collected in coagulant tubes. The tubes were gently shaken after blood collection and centrifuged at 3,000 rpm for 10 min at room temperature. The supernatant (serum) was collected in 1.5 mL frozen tubes and stored at −80°C for further analyses. After thawing on ice, metabolites in the serum samples were extracted using 50% methanol buffer. Briefly, 120 mL of precooled 50% methanol was added to 20 mL

of sample, vortexed for 1 min, incubated at room temperature for 10 min, and thereafter at −20°C, overnight. After centrifugation at 4,000 g for 20 min, the supernatants were transferred into new 96-well plates. quality control (QC) samples were prepared by pooling together 10 mL of each extract. The metabolites were stored at −80°C prior to the liquid chromatography-mass spectrometry (LC-MS) analysis. The samples were analyzed using a TripleTOF 5600 Plus high-resolution tandem mass spectrometer (Boston, USA) with both positive and negative ion modes. Chromatographic separation was performed using an ultra-performance liquid chromatography (UPLC) system (Boston, USA). Reversed-phase separation was performed using an ACQUITY UPLC T3 column (100 mm*2.1 mm, 1.8 mm) (Boston, USA). Eluted metabolites were detected and quantified using the TripleTOF 5600 Plus system. For the positive-ion mode, the ion spray floating voltage was set at 5 kV, whereas for the negative-ion mode, the voltage was set at −4.5 kV. The MS data were acquired in information-dependent acquisition (IDA) mode. The time-of-flight (TOF) mass range was 60–1,200 Da. During the entire period, the mass accuracy was calibrated after every 20 samples. Furthermore, the QC sample was analyzed after every 10 samples to evaluate the stability of LC-MS. Processing of the MS data including peak picking, peak grouping, retention time correction, second peak grouping, and annotation of isotopes and adducts was performed using XCMS software. LC-MS raw data files were converted into mzXML format before processing using XCMS, CAMERA, and metaX toolbox in R software.

## Hematoxylin-eosin (HE) and immunofluorescence detection of intestinal barrier

The proximal 1 cm colon of mice was isolated and washed three times with PBS for 5 min each, followed by fixation with 4% tissue fixative solution for 24 h. Then, paraffin embedding, sectioning, and staining were performed. Tight junction protein staining was detected by immunofluorescence.

## Western blotting analysis

Tissue and cell extracts were homogenized in ice-cold Radio-Immunoprecipitation Assay (RIPA) buffer with protease and phosphatase inhibitors and resolved on 8%–10% sodium dodecyl sulfate polyacrylamide gel electrophoresis (SDS–PAGE) gels and transferred to a polyvinylidene difluoride (PVDF) membrane. After blocking with 5% non-fat milk, the PVDF membrane was incubated with the desired primary antibody (diluted in TBST supplemented with 5% BSA) overnight at 4°C on an orbital shaker with gentle shaking, followed by rinsing with TBST three times, 8 min each at room temperature, and then, the secondary antibodies were incubated for 2 h at room temperature with gentle shaking. The secondary antibody (diluted in TBST supplemented with 5% BSA) was then removed, and the PVDF membrane was further washed three times with TBST for 8 min each at room temperature. Western blotting analysis was performed according to standard procedures.

## Transmission electron microscopy (TEM) staining

Collected MC3T3-E1 cells or colon tissue were fixed with 2.5% glutaraldehyde for 1 h at room temperature, followed by fixation with osmium tetroxide and embedding in Spurr's Epon. Ultra-thin sections (60 nm) were then cut and stained with uranyl acetate for 3 min. Then, operation was performed according to the electron microscope specimen instructions. The sections were observed with a Hitachi 7500 electron microscope and photographed with a digital camera.

## Statistical analysis

Statistical analyses were performed using SPSS 23.0 statistical software and R version 3.5.2. Continuous variables were expressed as mean ± standard deviation (SD).

Comparison between the two groups was performed using independent samples $t$-test; $\chi^2$ test was used to compare the differences between unordered categorical variables, and the rank-sum test was used to compare the differences between ordered categorical variables. Pearson correlation was used to show the relations between parameters of gut microbiota and serum metabolites. Statistical results were considered significant if $P$ values were < 0.05.

## RESULTS

### Participant demographics and clinical data

A total of 79 subjects with knee OA were recruited in the study, and the diagnosis of the disease is based on Kellgren and Lawrence grade and joint space narrowing (18). Thirty-nine subjects in the experimental group drank coffee, about 300 mg per day, at least 5 days per week, and 40 subjects in the control group never drank coffee. The two groups were matched for age, gender, body mass index (BMI), Kellgren and Lawrence grade, joint space narrowing, and complications. We did not find differences in BMD in medial tibial subchondral bone based on clinical quantitative computed tomography (QCT) analysis between the two groups ($P > 0.05$). In terms of markers of bone turnover, there were no significant differences in serum calcium, phosphate, and β-CTX between the two groups ($P > 0.05$). However, the levels of serum 25(OH)D$_3$, osteocalcin (OCN), alkaline phosphatase (ALP) and bone alkaline phosphatase (BALP) decreased significantly in the coffee group compared with the non-coffee group ($P < 0.05$) (Table 1).

### *P. Copri* abundance is associated with BMD of tibial subchondral bone in coffee-consumption subjects

To elucidate the potential difference in gut microbiota between coffee-consumption subjects and non-coffee-consumption subjects, shotgun metagenomic sequencing analyses of fecal samples were performed in this study. First, principal coordinate analysis (PCoA) analysis showed significantly different clustering of gut microbiota in coffee-consumption subjects compared with non-coffee-consumption subjects ($P < 0.01$, Fig. 1A). Top 30 significantly altered species ($P < 0.05$) between the coffee group and the non-coffee group were exhibited in Fig. 1B. Additionally, a significant difference in top

**TABLE 1** Comparison of general information between the coffee and non-coffee groups[a]

| Group | Coffee ($n = 39$) | non-Coffee ($n = 40$) | $\chi^2$ or $t$ | $P$ |
|---|---|---|---|---|
| Age (year) | 52.72 ± 5.42 | 52.15 ± 5.64 | 0.456 | 0.65 |
| Gender (man/women) | 11/28 | 13/27 | 0.172 | 0.678 |
| BMI (kg/m$^2$) | 22.61 ± 3.20 | 22.66 ± 3.66 | −0.072 | 0.942 |
| Diabetes | 5 | 6 | 0.078 | 0.780 |
| Hypertension | 7 | 7 | 0.003 | 0.958 |
| Cardiovascular disease | 6 | 7 | 0.064 | 0.800 |
| Hyper lipoprotein emia | 6 | 5 | 0.137 | 0.711 |
| BMD (medial tibial subchondral bone) (g/cm$^2$) | 0.833 ± 0.092 | 0.834 ± 0.091 | −0.056 | 0.955 |
| Kellgren and Lawrence Grade (0–4) | 3/7/16/11/2 | 4/7/20/7/2 | −0.750 | 0.454 |
| Joint space narrowing (0–3) | 4/18/14/3 | 4/16/16/4 | −0.527 | 0.598 |
| Calcium (mmol/L) | 2.29 ± 0.21 | 2.28 ± 0.21 | 0.045 | 0.964 |
| Phosphate (mmol/L) | 1.28 ± 0.19 | 1.29 ± 0.19 | −0.364 | 0.716 |
| 25(OH)D$_3$ (ng/mL) | 20.33 ± 3.68 | 26.53 ± 3.85 | −7.304 | <0.000 |
| Osteocalcin (µg/L) | 13.20 ± 4.24 | 18.16 ± 2.91 | −6.082 | <0.000 |
| ALP (U/L) | 34.38 ± 8.73 | 47.13 ± 8.20 | −6.689 | <0.000 |
| BALP (U/L) | 22.33 ± 3.77 | 32.58 ± 8.68 | −4.232 | <0.000 |
| β-CTX (pg/mL) | 676.21 ± 50.89 | 658.93 ± 29.46 | 1.853 | 0.068 |

[a]Data in the table are presented as mean±standard deviation (SD) or examples.

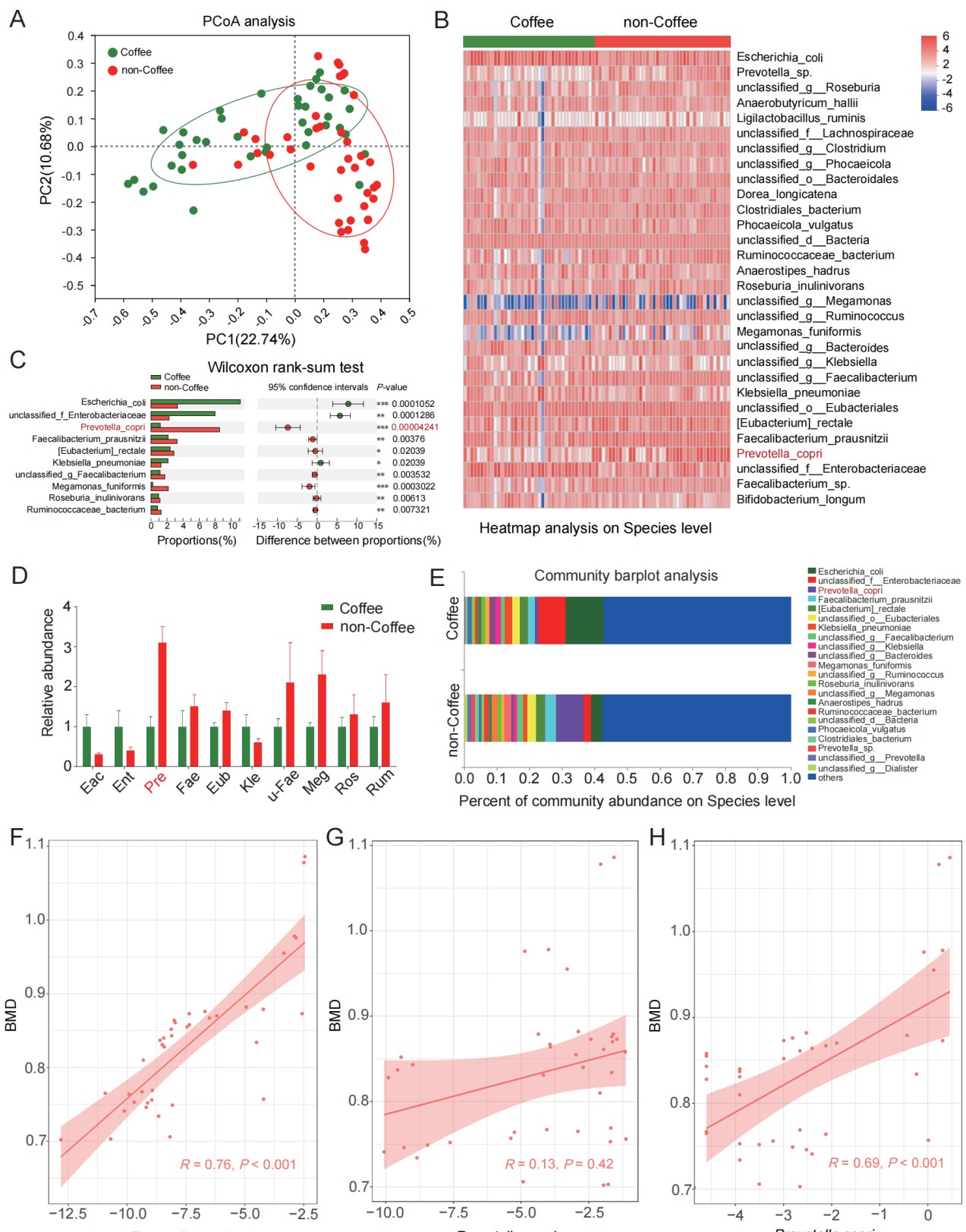

FIG 1 Composition differences in gut microbiota between coffee-consumption subjects (coffee) and non-coffee-consumption subjects (non-coffee). (A) Significance of PCoA analysis was accessed using Reads per Kilobase per Million mapped reads (RPKM) between the coffee group and the non-coffee group. (B) Top 30 significantly altered species between the two groups. Differences in abundance were detected by using RPKM. (C) Significant difference in the top 10

**Fig 1 (Continued)**

bacteria was calculated by Wilcoxon rank-sum test between the two groups. (D) The abundance of top 10 bacteria between the two groups was validated by using qPCR. Eac (*Escherichia_coli*), Ent (*unclassified_f_Enterobacteriaceae*), Pre (*Prevotella_copri*), Fae (*Faecalibacterium_prausnitzii*), Eub [(*Eubacterium*)_*rectale*], Kle (*Klebsiella_pneumoniae*), u-Fae (*unclassified_g_Faecalibacterium*), Meg (*Megamonas_funiformis*), Ros (*Roseburia_inulinivorans*), and Rum (*Ruminococcaceae_bacterium*). (E) Bacterial taxonomic profiling between the two groups at the species level. Correlative analysis of the abundance of *P. copri* and BMD of tibial subchondral bone in coffee-consumption subjects (F) or non-coffee-consumption subjects (G). (H) Correlation analysis of *P. copri* values was evaluated by using qPCR and BMD in coffee-consumption subjects. Correlations between variables were assessed by linear regression analysis. Linear correction index $R$ and $P$ values were calculated. *$P < 0.05$, **$P < 0.01$, ***$P < 0.001$.

10 bacteria calculated by Wilcoxon rank-sum test was shown in Fig. 1C between the two groups. At the same time, the differential top 10 bacteria were confirmed by quantitative PCR (qPCR) (Fig. 1D). These results showed that *P. copri* has a maximum multiple of difference not only in shotgun metagenomic sequencing analyses but also in qPCR results between the two groups. Additionally, the percent of community abundance on the species level between the two groups is shown in Fig. 1E.

Next, the relationship between *P. copri* abundance and BMD of tibial subchondral bone was evaluated. In coffee-consumption subjects, *P. copri* abundance evaluated according to the results of shotgun metagenomic sequencing analyses was positively correlated with BMD ($P < 0.001$, Fig. 1F), but we did not observe a correlation between them in non-coffee-consumption subjects ($P = 0.42$, Fig. 1G). Additionally, a similar correlation was shown between *P. copri* abundance evaluated by using the qPCR values and BMD in coffee-consumption subjects ($P < 0.001$, Fig. 1H). Taken together, *P. copri* has a great correlation with BMD of tibial subchondral bone in coffee-consumption subjects with OA.

## Caffeine metabolism and BMD of tibial subchondral bone in coffee consumption OA subjects

Serum metabolomics tests were determined by using LC-MS analysis for the subjects. Partial least squares discriminant analysis (PLS-DA) showed that serum metabolic profiles in coffee-consumption subjects were significantly different from non-coffee-consumption subjects (Fig. 2A). Top 50 significantly altered metabolites ($P < 0.05$) between coffee group and non-coffee group were exhibited in Fig. 2B. Caffeine metabolism was the top enriched pathway in coffee-consumption subjects compared with non-coffee-consumption subjects (Fig. 2C). In human body, paraxanthine, theobromine, and theophylline are the mainly metabolized substances of caffeine (19). The concentration of the above caffeine-related metabolites in serum was further detected by targeted mass spectrometry (targeted MS) and showed the same variation tendency between the two groups (Fig. 2D). In coffee-consumption subjects (all $P < 0.01$, Fig. 2E), the serum level of caffeine-related metabolites detected by LC-MS were significantly negatively correlated with BMD of tibial subchondral bone, but we did not observe a significantly similar correlation between them in non-coffee-consumption subjects (Fig. 2F). Additionally, there were significant correlations between caffeine-related metabolites detected by targeted MS and BMD in coffee-consumption subjects (all $P < 0.001$, Fig. 2G). Together, these results indicated that caffeine metabolism may be involved in BMD decrease of subcondral bone in OA subjects who consumed coffee.

## Correlation analysis of gut microbiota and caffeine-related metabolites

To determine the potential relevance of gut microbiota with serum caffeinerelated metabolites, correlation analysis was performed on the total 79 subjects. We observed that *P. copri* evaluated by shotgun metagenomic sequencing had the most negative correlation with caffeine concentration when evaluated by LC-MS (Fig. 3A). Correlation analysis showed that the level of caffeine in serum had a significant negative correlation with the abundance of *P. copri* in the feces in coffee-consumption subjects (Fig. 3B),

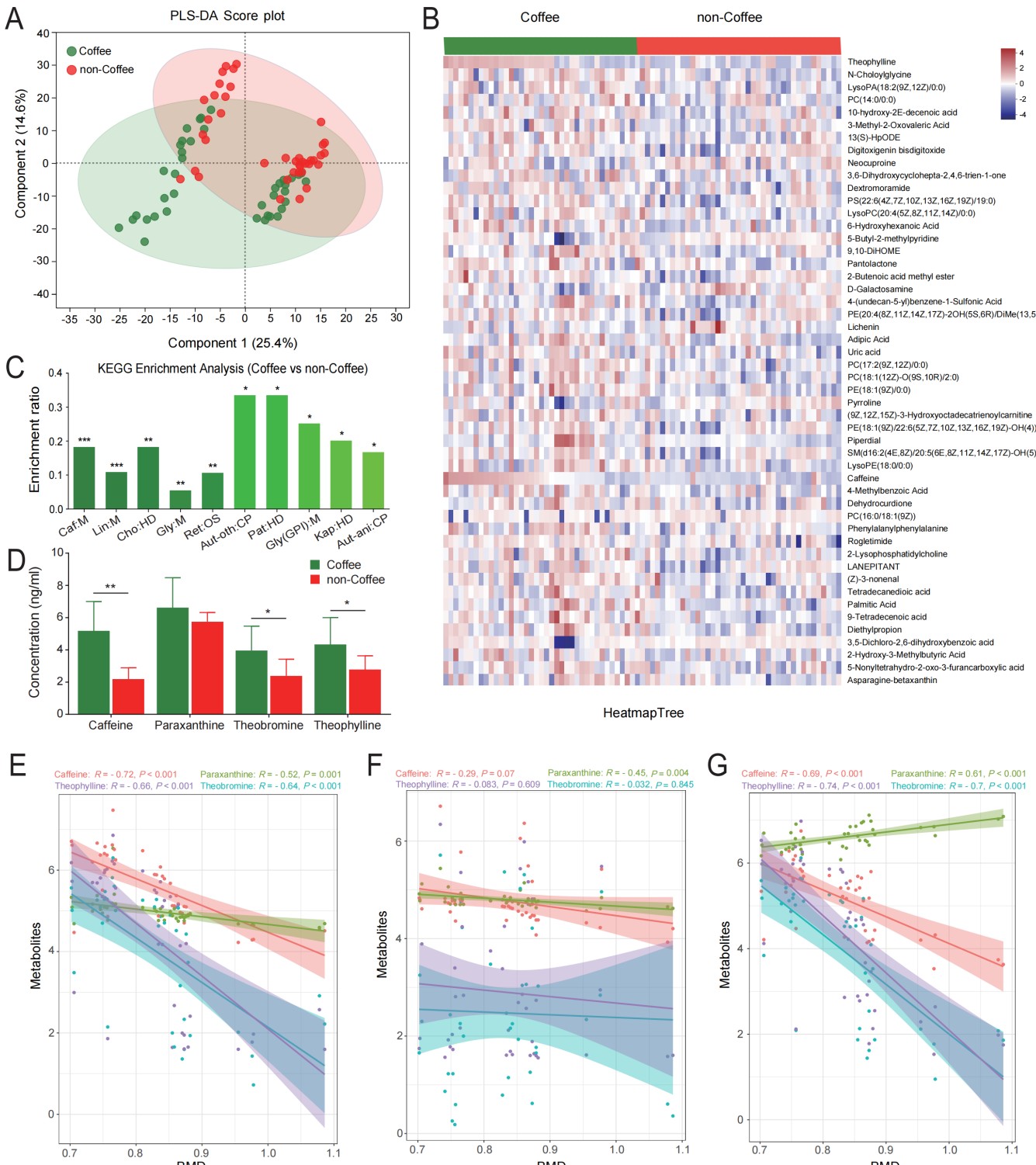

**FIG 2** Differences in serum metabolites between coffee-consumption subjects and non-coffee-consumption subjects. (A) Serum metabolic profile was significantly different between coffee group and non-coffee group by partial least squares discriminant analysis (PLS-DA) method, which is a supervised multiple regression analysis for identifying discernible patterns. (B) Differential metabolites between the two groups, two-tailed Mann-Whitney U test. (C) Kyoto Encyclopedia of Genes and Genomes (KEGG) enrichment analysis of differential metabolites between the two groups. Caf (Caffeine metabolism), Lin (Linoleic acid metabolism), Cho (Choline metabolism in cancer), Gly (Glycerophospholipid metabolism), Ret (Retrograde endocannabinoid signaling), Aut-oth (Autophagy-other), Pat (Pathogenic Escherichia coli infection), Gly(GPI) [Glycosylphosphatidylinositol (GPI)-anchor biosynthesis], Kap (Kaposi sarcoma-associated herpesvirus infection), and Aut-ani (Autophagy-animal). (D) The serum concentration of caffeine-related metabolites (paraxanthine, theobromine, and

Fig 2 (Continued)

theophylline) between the two groups was measured by targeted MS assay, two-tailed Student's *t*-test. Correlation analysis of BMD and serum concentration of caffeine-related metabolites detected by LC-MS in coffee-consumption subjects (E) or non-coffee-consumption subjects (F). (G) Correlation between BMD and caffeine detected by targeted MS in coffee-consumption subjects. Correlations between variables were assessed by linear regression analysis. Linear correction index *R* and *P* values were calculated. *$P < 0.05$, **$P < 0.01$, ***$P < 0.001$.

whereas no significant correlation was observed between them in non-coffee-consumption subjects (Fig. 3C). In keeping with this, we further confirmed that the abundance of *P. copri* in feces evaluated by qPCR was negatively correlated with the concentration of caffeine in serum detected by targeted MS (Fig. 3D). Thus, there was a close correlation between dysbiosis of *P. copri* in feces and caffeine level in serum in coffee-consumption OA subjects. Next, the relationships among *P. copri* in feces evaluated by qPCR, serum caffeine-related metabolites detected by targeted MS, and serum bone-related metabolic indicators detected by targeted MS were examined by correlation analysis (Fig. 3E). In terms of bone metabolic indicators, the abundance of *P. copri* in feces was found to be positively correlated to serum markers of osteoblast activity, such as osteocalcin (OCN), ALP, and BALP, whereas negatively correlated to serum markers of osteoclast

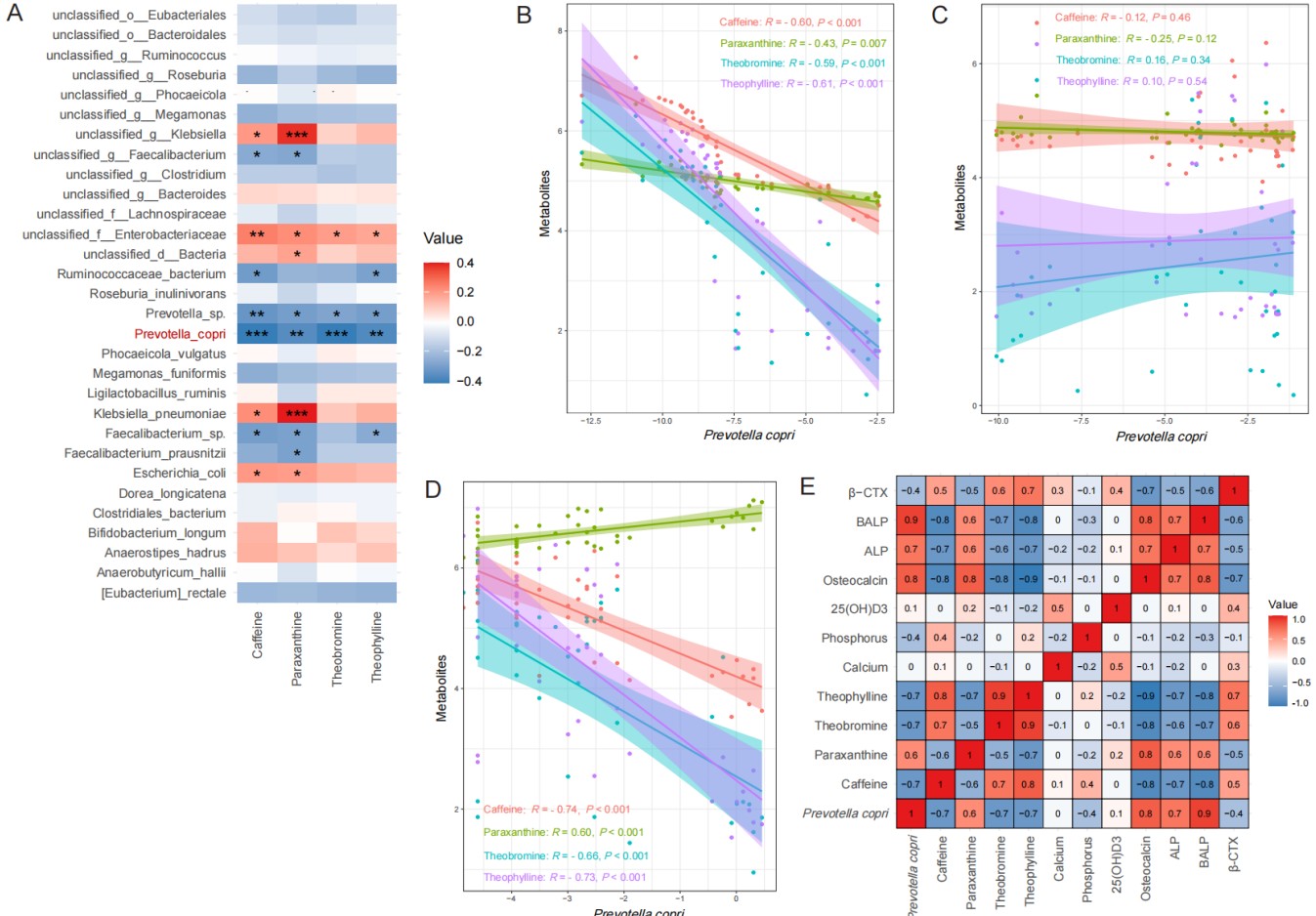

**FIG 3** Correlation analysis diagram. (A) Heatmap of correlations between the significant top 30 fecal species and serum caffeine-related metabolites in 79 subjects. Red indicates a strong positive correlation, and blue indicates a strong negative correlation. Correlative analysis of *P. copri* in feces was evaluated by shotgun metagenomic sequencing, and serum caffeine-related metabolites were detected by LC-MS in coffee-consumption subjects (B) or non-coffee-consumption subjects (C). (D) Correlation between *P. copri* values evaluated by using qPCR and the level of caffeine-related metabolites in serum detected by targeted MS. (E) The correlation among *P. copri* value in feces evaluated by qPCR, caffeine-related metabolites, and bone-related indicators in serum detected by targeted MS. Correlation analyses were performed using pearson correlation coefficient. *$P < 0.05$, **$P < 0.01$, ***$P < 0.001$.

activity, such as β-CTX, and caffeine in serum was inversely positively correlated. These results suggested that *P. copri* may mediate caffeine degradation, and the function potentially safeguards BMD in OA subjects who consumed coffee.

## Caffeine intake aggravates subchondral bone loss and impairs gut barrier function of colon in DMM mice

To explore the effect of caffeinerelated metabolites on OA, DMM mice were intervened by caffeine-related metabolite intake. In caffeine intake group, there was a decreased tibial subchondral bone mass analyzed by Micro-CT (Fig. 4A through F) and aggravated articular cartilage destruction displayed by safranin O-fast green staining (Fig. 4G and H); however, the subchondral bone mass increased, and articular cartilage destruction relieved in paraxanthine-intake group, whereas no noteworthy changes were observed in theobromine- or theophylline-intake groups, compared with the DMM mice group. As for the immunofluorescence staining (Fig. 4I) and western blot analysis (Fig. 4J), the results suggested that the caffeine-intake group exhibited a significantly declined tendency of OCN, and paraxanthine-intake group exhibited an increased tendency, whereas no noteworthy changes were noted in theobromine- or theophylline-intake groups, compared with the DMM mice group. Furthermore, the serum level of OCN significantly decreased after caffeine intake and increased after paraxanthine intake, whereas no remarkable changes in theobromine- or theophylline-intake groups, relative to that in DMM mice group (Fig. 4K). Collectively, these data revealed that caffeine intake was able to accelerate, whereas paraxanthine intake could restore, the tibial subchondral bone loss in the mice with DMM-induced OA.

Next, in order to explore the effects of caffeine intake on the integrity of gut barrier in mice with DMM-induced OA, we conducted further exploration. The HE staining of colon (Fig. 4L) showed that the intestinal cavity of the mice with caffeine intake was relatively sparse, and the intestinal gap was significantly enlarged; paraxanthine-intake group showed a dense intestinal cavity and reductive intestinal gap, whereas theobromine- or theophylline-intake groups expressed an equivalent manifestation, compared with DMM mice group. Additionally, as demonstrated by double immunofluorescence staining (Fig. 4M) and western blot analysis (Fig. 4N), the expressions of claudin-3 and ZO-1, the tight junction proteins, in colon tissue were relatively low in caffeine-intake group, which may result in high intestinal permeability, compared with DMM mice group. However, in paraxanthine-intake group, the colon tissue exhibited a higher expression of claudin-3 and ZO-1, which may decrease intestinal permeability, compared with DMM mice group. Meanwhile, we did not find that the levels of claudin-3 and ZO-1 changed in colon tissue in theobromine- or theophylline-intake groups compared with DMM mice group. In general, these results suggested that caffeine intake can jeopardize the integrity of intestinal barrier and increase intestinal permeability, whereas paraxanthine intake was able to restore the integrity of intestinal barrier and decrease intestinal permeability.

In addition, we observed a decrease in fecal *P. copri* abundance in the caffeine-intake group, but no remarkable changes in paraxanthine-, theobromine-, or theophylline-intake groups, relative to the DMM mice group (Fig. 4O). Noteworthy, the decreased fecal *P. copri* abundance in the caffeine-intake mice is consistent with the coffee-consuming population in the present study. The above results suggested that the depletion of *P. copri* abundance may be involved in caffeine metabolism in DMM mice model.

In conclusion, these data establish intestinal caffeine accumulation may act as a risk factor, whereas paraxanthine accumulation could exhibit protective effects, for the subchondral bone mass in DMM mice. Meanwhile, the intestinal barrier may be involved in the above pathological process. Additionally, the result also suggested that *P. copri* may exhibit potential caffeine-degrading ability in DMM mice.

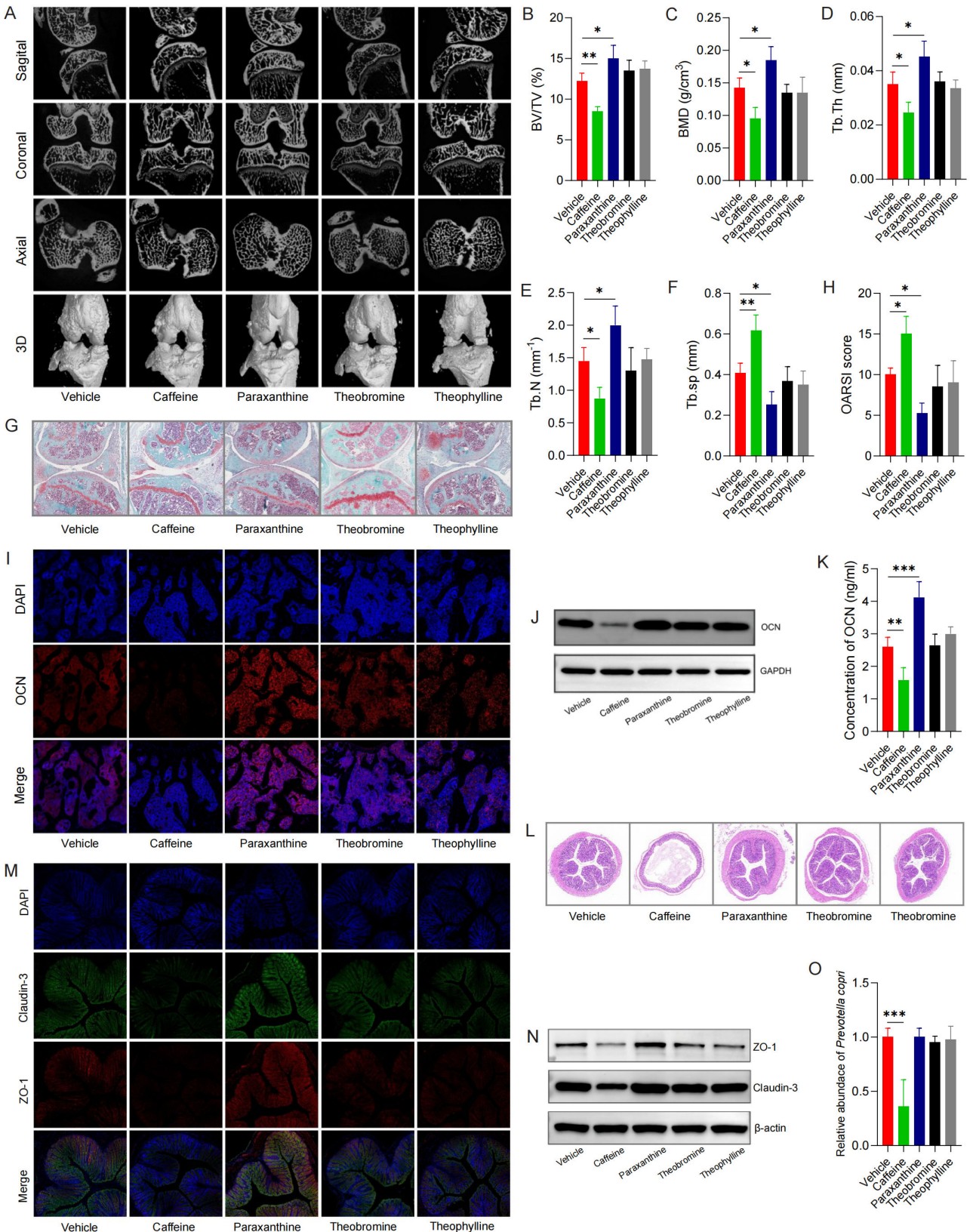

**FIG 4** Administration of caffeine-related metabolites on subchondral bone and gut barrier function in DMM mice model. (**A**) Representative micro-CT images for the three-dimensional (3D) reconstruction of knee joints with axial, sagittal, and coronal views of the subchondral bone (*n* = 4 for each group). (**B-F**) Quantitative analysis of micro-CT parameters of tibial subchondral bone: (B) bone volume/tissue volume (BV/TV), (C) bone mineral density (BMD), (D) trabecular thickness (Continued on next page)

Fig 4 (Continued)

(Tb.Th), (E) trabecular number (Tb.N), and (F) trabecular separation (Tb.sp). Safranin O-fast green staining (G) and Osteoarthritis Research Society International (OARSI) score (H) of the medial tibial articular cartilage. Scale bar = 200 µm. (I) The immunofluorescence staining of OCN in subchondral bone. Scale bar = 50 µm. (J) Protein levels of OCN of subchondral bone detected by western blotting. (K) The serum concentration of OCN among groups. (L) Representative HE staining of the colonic tissues among groups. Scale bar = 200 µm. (M) Protein expression of claudin-3 and ZO-1 in the colon tissue by double immunofluorescence staining. Claudin-3 (green), ZO-1 (red), and DAPI (blue), Scale bar = 50 µm; (N) Protein expression of claudin-3 and ZO-1 in the colon tissue by western blotting. (O) Fecal *P. copri* abundance was detected by using qPCR. Data were expressed as the mean ± SD. One-way ANOVA procedures were used to assess statistical significance. *$P < 0.05$, **$P < 0.01$, ***$P < 0.001$.

## Caffeine or paraxanthine interference and ferroptosis of MC3T3-E1 cells

According to previous reports, various caffeine concentrations were employed separately to determine the optimal inducing concentration (Fig. 5A). We found that caffeine interference decreased MC3T3-E1 viability, in a time- and dose-dependent manner. After 24 h of treatment with 20 µM caffeine, MC3T3-E1 viability was significantly reduced. When higher caffeine concentration was affiliated, not enough MC3T3-E1 cells could be collected to perform the subsequent experiments. Therefore, a regimen of caffeine at 20 µM for 24 h was used in this study to facilitate the observation of MC3T3-E1 death. To study the relative contributions of ferroptosis to caffeine-induced MC3T3-E1 death, we evaluated the rescue effects of ferroptosis inhibitors, Fer-1(10 µM) and iron chelator desferrioxamine (DFO, 200 µM), using a CCK-8 assay (Fig. 5B) (20, 21). The results showed that both the two ferroptosis inhibitors, especially Fer-1, significantly rescued caffeine-induced MC3T3-E1 death.

To further confirm the occurrence of ferroptosis by caffeine interference, we performed double immunofluorescent labeling (Fig. 5C) and western blotting analysis (Fig. 5D) in caffeine interference MC3T3-E1 with or without Fer-1 rescue. Consistent with the CCK-8 assay, both the double immunofluorescent labeling assay and western blotting analysis confirmed that caffeine-interfering MC3T3-E1 death could be significantly reduced by Fer-1 treatment.

Additionally, we found that paraxanthine increased MC3T3-E1 viability in erastin (25 µM) pretreated cells in a dose-dependent manner exceeding 5 µM (Fig. 5E) (22). In erastin-pretreated MC3T3-E1 cells, we found that paraxanthine significantly relieved ferroptosis of MC3T3-E1 cells both at the double immunofluorescent labeling assay (Fig. 5F) and western blotting analysis (Fig. 5G).

Taken together, the above results demonstrated that caffeine interference facilitated ferroptosis, whereas paraxanthine interference restrained in MC3T3-E1 cells.

## *P. copri* treatment restores caffeine-induced ferroptosis of osteoblast *in vitro* and *in vivo*

*P. copri* treatment decreased MC3T3-E1 viability compared with that of the control group in a time- and dose-dependent manner. After 24 h of treatment with $10^8$ CFU/mL of *P. copri*, MTT result showed MC3T3-E1 viability was reduced to 77% (Fig. 6A). When higher *P. copri* concentrations were used, not enough MC3T3-E1 cells could be collected to perform the subsequent experiments. Herein, a concentration of *P. copri* at $10^8$ CFU/mL for 24 h interference was used to facilitate the observation of MC3T3-E1 death. In caffeine interference MC3T3-E1 cells, *P. copri* treatment significantly increased the expression levels of OCN and GPX4 by fluorescence intensities and western blotting analysis (Fig. 6B and C). Moreover, the ultrastructural analysis from TEM observation (Fig. 6D) showed that the mitochondria in the caffeine interference group were swollen with blurred ridges, but the mitochondria morphology became normal in the *P. copri* treatment group. Additionally, in caffeine interference MC3T3-E1 cells, paraxanthine level increased, whereas caffeine level decreased, after 24 h treatment of *P. copri* in a dose-dependent manner (Fig. 6E). These results suggested that *P. copri*-mediated caffeine

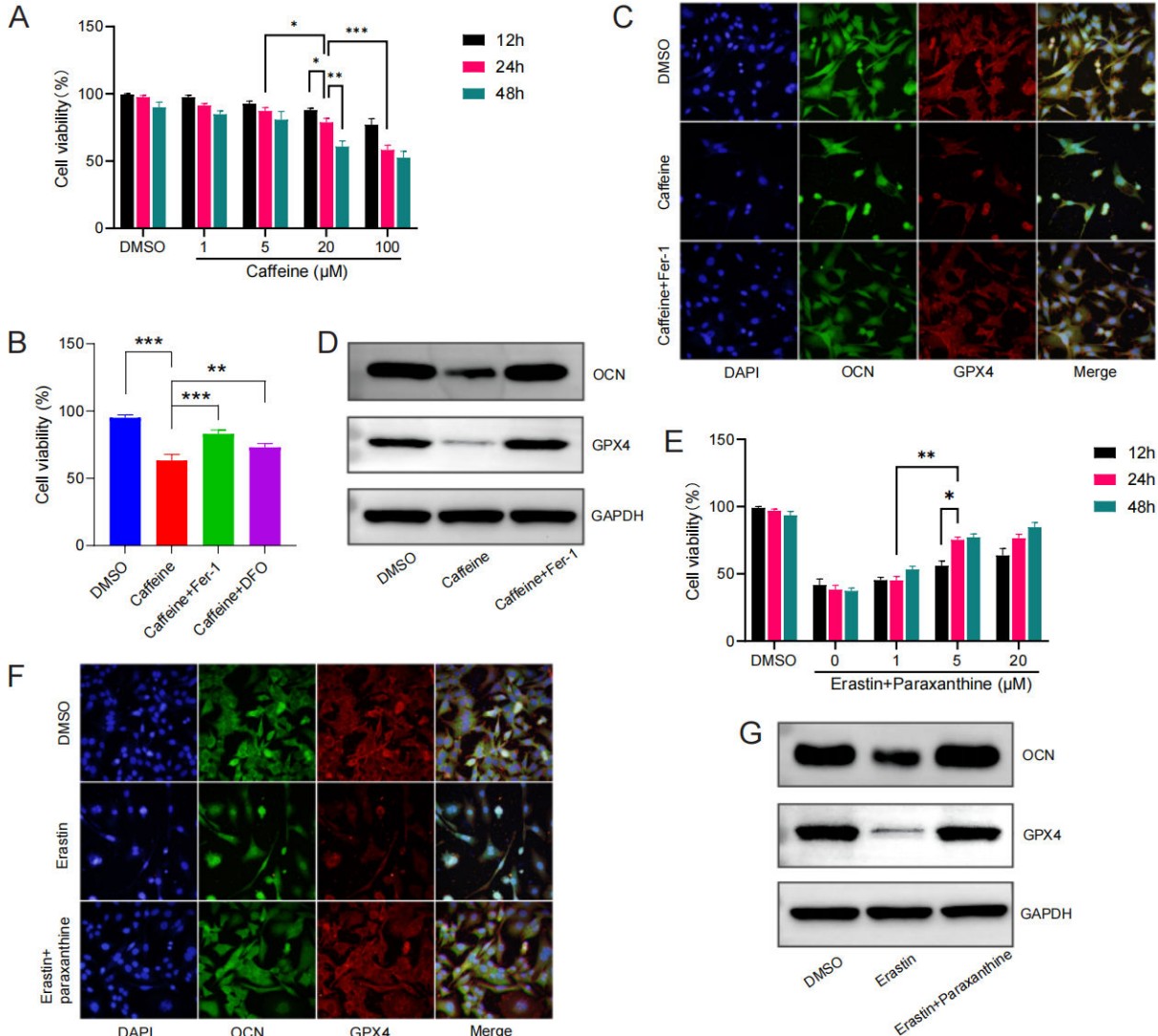

**FIG 5** Induction of ferroptosis in MC3T3-E1 cells by caffeine or paraxanthine. (A) MC3T3-E1 cells were treated with dimethyl sulfoxide (DMSO) (solvent) plus various concentrations of caffeine (0, 1, 5, 20, and 100 µM) at different times and evaluated by CCK-8 assay. (B) MC3T3-E1 cells pretreated with DMSO, Fer-1 (10 µM) (a ferroptosis inhibitor), iron chelator desferrioxamine (DFO) (200 µM) (a specific ferroptosis inhibitor), and subjected to caffeine treatment for 24 h, and then evaluated by CCK-8 assay. (C) MC3T3-E1 cells were treated with DMSO, caffeine, or caffeine +Fer-1 for 24 h. The expression of OCN and GPX4 were examined using double immunofluorescent labeling. OCN (green), GPX4 (red), and DAPI (blue), Scale bar = 20 µm. (D) Western blotting was used to assess the protein levels of OCN and GPX4 in MC3T3-E1 cells after caffeine interference with or without Fer-1 rescue. (E) MC3T3-E1 cells were pretreated with erastin (25 µM) (a ferroptosis inducer) for 24 h, and then treated with DMSO plus various concentrations of paraxanthine (0, 1, 5, and 20 µM) for different times and evaluated by CCK-8 assay. The expression of OCN and GPX4 in MC3T3-E1 cells were examined using double immunofluorescent labeling (F) and western blotting (G) in erastin pretreated for 24 h and then treated with paraxanthine (5 µM) interference for 24 h. OCN (green), GPX4 (red), and DAPI (blue), Scale bar = 20 µm. *$P < 0.05$, **$P < 0.01$, ***$P < 0.001$.

degradation and alleviated caffeine-induced ferroptosis by promoting the breakdown of caffeine into paraxanthine in MC3T3-E1 cells.

To elucidate whether caffeine-induced subchondral bone loss could be alleviated by colonizing *P. copri* in the gut, *P. copri* suspension was transplanted to the recipient mice by gavage. DMM mice were administered with PBS (vehicle), caffeine water, or *P. copri* ($10^8$ CFU/mL) plus caffeine water. The effects of *P. copri* treatment on tibial subchondral bone mass were analyzed by micro-CT. At the same time, we evaluated the role of *P. copri* treatment in the caffeine-induced ferroptosis of osteoblasts, which are responsible for subchondral bone formation. The results of micro-CT images (Fig. 6F through K) and

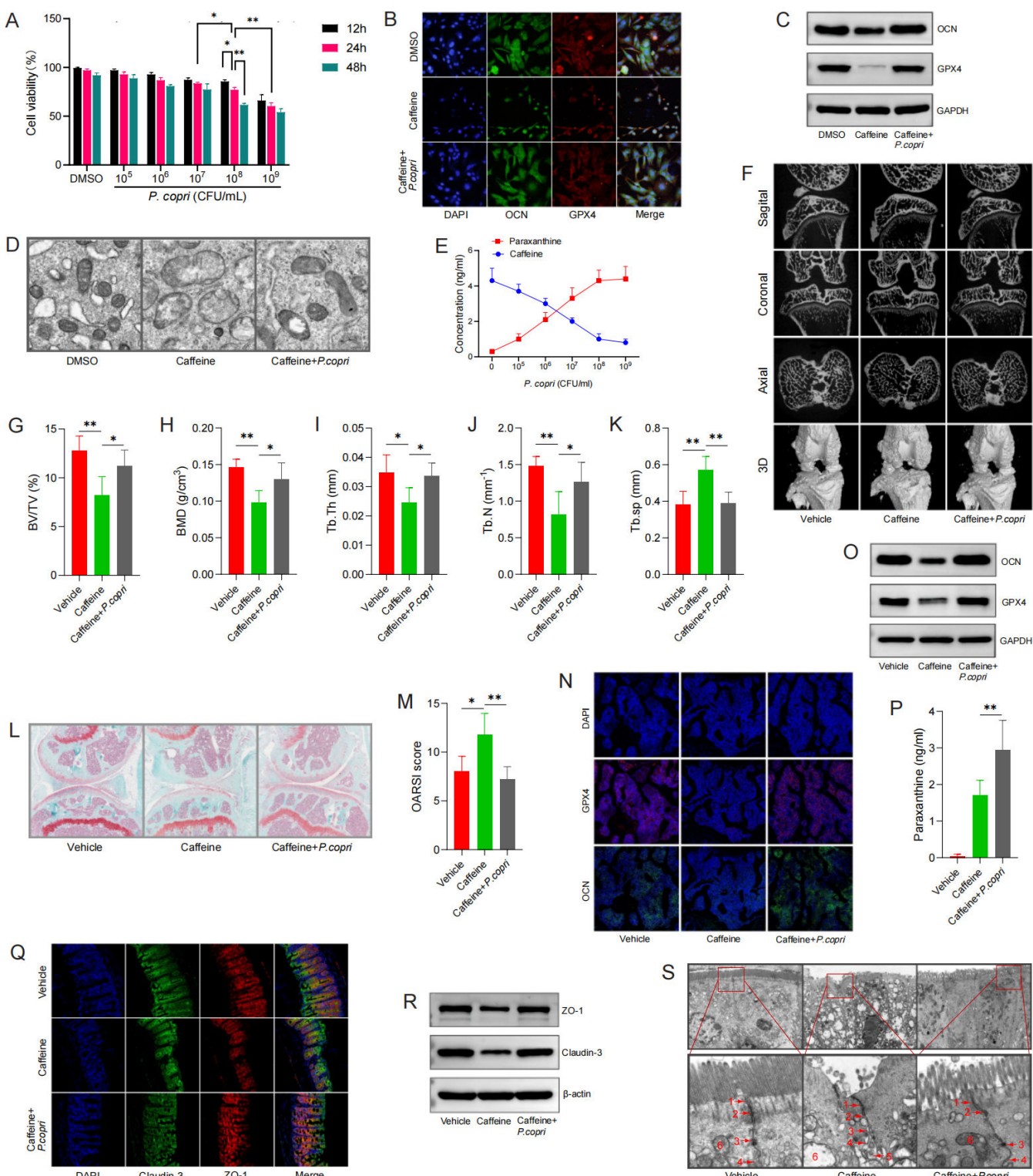

**FIG 6** *P. copri* alleviated caffeine-induced ferroptosis *in vitro* and *in vivo*. (A) MC3T3-E1 cells were treated with DMSO plus various concentrations of *P. copri* (0, $10^5$, $10^6$, $10^7$, $10^8$, or $10^9$ CFU/mL) for different times and evaluated with a CCK-8 assay. (B) The expression of OCN and GPX4 were examined using double immunofluorescent labeling in caffeine (20 μM) interference MC3T3-E1 cells with or without *P. copri* treatment. OCN (green), GPX4 (red), and DAPI (blue). Scale bar = 20 μm. (C) Western blotting was used to assess the protein levels of OCN and GPX4 in caffeine interference MC3T3-E1 cells with or without *P. copri* ($10^8$ CFU/mL) treatment. (D) Mitochondrial ultrastructure of MC3T3-E1 cells under the indicated treatments by transmission electron microscopy (TEM). Scale bar = 2 μm. (E) Paraxanthine expression increased after various concentrations of *P. copri* treatment detected by targeted MS in caffeine (20 μM) interferenced (Continued on next page)

**Fig 6 (Continued)**

MC3T3-E1 cells. (F) Representative micro-CT images of knee joints with axial, sagittal, and coronal, and the 3-dimensional (3D) reconstruction views of tibial subchondral bone ($n = 5$ for each group). (G-K) Quantitative analysis of micro-CT parameters of tibial subchondral bone: (G) bone volume/tissue volume (BV/TV), (H) bone mineral density (BMD), (I) trabecular thickness (Tb.Th), (J) trabecular number (Tb.N), and (K) trabecular separation (Tb.sp). Safranin O-fast green staining (L) and Osteoarthritis Research Society International (OARSI) score (M) of the medial tibial articular cartilage. Scale bar = 200 µm. The expression of OCN and GPX4 in subchondral bone were examined using immunofluorescent labeling (N) and western blotting (O) in caffeine intake DMM mice with or without *P. copri* treatment. OCN (green), GPX4(red), and DAPI (blue). Scale bar = 50 µm. (P) Paraxanthine expression in serum increased after *P. copri* treatment detected by targeted MS in caffeine intake DMM mice. Double immunofluorescence (Q) staining and western blotting (R) of claudin-3 and ZO-1 in colon of the DMM mice. Claudin-3 (green), ZO-1 (red), and DAPI (blue), Scale bar = 20 µm. (S) The representative images of colon observed by TEM, Scale bar = 1 µm. (1) Tight junction, (2) adherens junction, (3) desmosome, (4) gap junction, (5) disrupted cell junction, (6) and mitochondria. *$P < 0.05$, **$P < 0.01$, ***$P < 0.001$.

safranin O-fast green staining (Fig. 6L and M) showed that caffeine-induced subchondral bone loss and articular cartilage destruction was reversed after colonizing *P. copri* in the gut of DMM mice. Additionally, the expressions of OCN and GPX4 in tibial subchondral bone were also reversed after colonizing *P. copri* in gut in caffeine intake DMM mice, as shown by fluorescence staining (Fig. 6N) and western blotting analysis (Fig. 6O). Importantly, serum paraxanthine concentrations were significantly increased after *P. copri* treatment, compared with caffeine intake DMM mice (Fig. 6P). These results suggested that *P. copri* treatment could restore bone homeostasis and enhance the ferroptosis resistance of osteoblasts *in vitro* and *in vivo*.

To investigate the impact of colonizing *P. copri* on gut barrier function, we analyzed the expression levels of colon-tight junction proteins *in vivo*. Colonizing *P. copri* in the gut markedly reversed the decreased levels of claudin-3 and ZO-1 induced by caffeine intake as determined by double immunofluorescence staining (Fig. 6Q) and western blotting (Fig. 6R). Moreover, in the colon of caffeine intake DMM mice, TEM images showed an impaired tight junction, swollen mitochondria, and reduced cristae, but the deterioration of gut barrier was rescued after *P. copri* treatment (Fig. 6S). Taken together, these data suggested that colonizing *P. copri* in gut preserves the intestinal barrier impairment in the caffeine-interfering DMM mice.

## DISCUSSION

Here, we establish for the first time, a novel gut microbiota-mediated mechanism of caffeine-induced OA progression. Despite several studies showing an association between caffeine and OA, it remains unclear whether gut microbiota take part in the caffeine-induced subchondral bone loss and articular cartilage degeneration (3, 23–25). To fill these critical gaps, we recruited volunteers to explore the aberrance of fecal gut microbiota and serum metabolites in OA subjects with coffee consumption and performed experiments involving the effect of caffeine intake on the ferroptosis of osteoblasts *in vitro* and *in vivo*.

In this study, we demonstrated that the structure of gut microbiota was disturbed in OA subjects with long-term coffee consumption. Particularly, the abundance of intestinal *P. copri* was significantly exhausted in the coffee consumption OA subjects, and the decreased level of it has a negative correlation with subchondral bone mass. *P. copri* is an abundant member of the human gastrointestinal microbiome, whose relative abundance curiously has been associated with positive and negative impacts on several diseases (26). The abundance of *P. copri* is facilitated and affected by diet, lifestyle, and geographical location (27, 28). Therefore, there is a potential correlation between intestinal *P. copri* abundance and the OA progression in coffee consumers.

After caffeine is ingested in the human body, it is decomposed into paraxanthine, theobromine, or theophylline in the liver by enzymes; however, there has been debate for a long time on how their distributions of multiple products are determined (29). Herein, the data showed that there was a negative correlation between serum caffeine level and bone mass in coffee consumers, which is consistent with the results of several previous studies (30, 31). It has been reported that gut microbiota are involved in

caffeine catabolism (32). For this reason, we analyzed the correlation between gut microbiota and serum caffeine-related metabolites in the subjects. The results showed that the abundance of *P. copri* in the gut was most closely related to the content of serum caffeine, and there was a negative correlation between them in coffee consumers. Gut microbiota affects the metabolic rate of intestinal substances and is an important regulator for the progression of diseases (33, 34). Chen et al showed that *Bacteroides xylanisolvens* was a key contributor to nicotine metabolism in smokers (35). Newman et al reported that *P. copri* appears to mediate metabolic dysfunction in western diet-fed non-human primates (36). Together, the above evidence gave a new clue that *P. copri* may affect caffeine metabolism.

Consistent with the results of this study, it has been reported that there is a negative correlation between serum caffeine and subchondral bone mass, but it is not clear whether there is a causal relationship between them (11). At the same time, in an analysis included a total of 1,235 adolescents, the data showed that there is a positive association between urinary paraxanthine and BMD; however, there has not been any research on the relationship between serum paraxanthine and OA progression (12). Therefore, to fill the above gaps, we verified the effect of caffeine-related metabolites on subchondral bone in the DMM-induced mice model. The data established intestinal caffeine accumulation may act as a risk factor, whereas paraxanthine accumulation could exhibit protective effects for the progression of OA. Furthermore, the results also showed that caffeine intake can jeopardize the integrity of intestinal barrier and increase the intestinal permeability, whereas paraxanthine intake was able to restore the integrity of the intestinal barrier and decrease intestinal permeability. In addition, we observed a remarkable decrease of fecal *P. copri* abundance and an increase in serum paraxanthine concentration in caffeine-intake DMM mice. Noteworthy, the decreased fecal *P. copri* abundance in the caffeine-intake mice is consistent with the coffee-consuming population in the present study. Together, the above results suggested that depletion of *P. copri* abundance in gut may be involved in caffeine metabolism.

Ferroptosis is characterized by the overgeneration of lipid peroxidation in response to the downregulation of GPX4 (37). This programmed cell death is involved in the development of osteoblast dysfunctions, whereas GPX4 level was downregulated (38). Therefore, targeting ferroptosis of osteoblast is a potential treatment approach for BMD variation (39). The present results demonstrated that caffeine interference facilitated ferroptosis, whereas paraxanthine interference restrained in MC3T3-E1 cells. Here, human samples and mice experiments have shown that *P. copri* may have a potential prompting role in caffeine metabolism. Nii et al revealed that in bone marrow-derived dendritic cells of rheumatoid arthritis model, *P. copri* have the effect to induce a strong innate immune response (40). Therefore, in order to verify the ability of caffeine degrading, we conducted further exploration *in vitro*, the data suggested that *P. copri* could mediate caffeine degradation and may alleviate caffeine-induced ferroptosis by promoting the breakdown of caffeine into paraxanthine in MC3T3-E1 cells. Additionally, we found that caffeine-induced subchondral bone loss and ferroptosis of osteoblast recovered after intestinal *P. copri* implantation in DMM mice. Importantly, serum paraxanthine concentrations were significantly increased after intestinal *P. copri* colonizing, compared with the DMM mice just with caffeine intake. These results suggested that *P. copri* colonizing could restore bone loss and enhance ferroptosis resistance of osteoblasts *in vitro* and *in vivo*. Under normal circumstances, the paracellular gaps are sealed by tight junctions to maintain the integrity and the functions of gut barriers (41, 42). Intestinal barrier impairment is a clear histopathology manifestation, and intervention based on the recuperation of the intestinal barrier is an effective means for OA treatment (43, 44). The present data showed that colonizing *P. copri* in gut preserves the intestinal barrier impairment in the caffeine-induced DMM mice. Therefore, *P. copri*-mediated bone mass recovery may be related to its impact on the repair of intestinal barrier.

## Conclusion

This study demonstrates for the first time that gut microbiota disturbance is involved in coffee consumption-mediated OA and that enrichment of *P. copri* in gut reversed tibial subchondral bone loss by promoting caffeine metabolism. The structure of gut microbiota was disturbed in OA subjects with long-term coffee consumption. For coffee consumers, *P. copri* abundance in gut was most closely related to bacteria and negatively correlated with the content of serum caffeine. Importantly, the decreased fecal *P. copri* abundance was also found in the caffeine intake mice. Then, we found that intestinal caffeine accumulation may act as a risk factor, whereas paraxanthine accumulation could exhibit protective effects for the progression of OA. *P. copri* colonizing could restore bone loss and enhance the ferroptosis resistance of osteoblasts *in vitro* and *in vivo*. Furthermore, we also demonstrated that *P. copri*-mediated bone mass recovery may be related to its impact on the repair of intestinal barrier. Therefore, for coffee consumers, intestinal *P. copri* supplementation may be a potential treatment method for alleviating the progression of OA. This study also gives us a new perspective that gut microbiota may be involved in the dietary regulation of OA progression. Hence, the scientific popularization of dietary regulation on OA is of great significance, which can be regarded as the most common intervention approach in daily life to participate in the prevention of OA. It suggests that gut microbiota might be one of the main exploration directions for basic research and clinical management of OA in the future.

However, there is a noticeable dispersion in the characteristics of gut microbiota and serum metabolites within the experimental groups. Further large-scale population-based cohorts are still needed to comprehensively assess the effects of demographic characteristics, dietary patterns, lifestyle, genetic background, and other factors on gut microbiota and serum metabolites in individual coffee consumers.

## ACKNOWLEDGMENTS

This study was supported by grants from National Natural Science Foundation of China (no. 81802164), Medical Scientific and Technological Research Project of Henan Province (no. 212102310109), Key Scientific Research Project of Education Department of Henan Province (no. 21A320023).

## AUTHOR AFFILIATIONS

[1]Department of Orthopedics, The First Affiliated Hospital of Zhengzhou University, Zhengzhou, China
[2]Department of Orthopedics, People's Hospital of Zhengzhou, Zhengzhou, China

## AUTHOR ORCIDs

Feng Li ⓘ http://orcid.org/0000-0002-1765-5359
Xianwei Wang ⓘ http://orcid.org/0009-0009-7383-0498
Guofu Pi ⓘ http://orcid.org/0000-0002-7815-4005

## AUTHOR CONTRIBUTIONS

Feng Li, Data curation, Formal analysis, Funding acquisition, Investigation, Writing – original draft, Software, Validation | Xin Wen, Data curation, Formal analysis, Investigation, Methodology, Validation, Software | Pu Xue, Data curation, Investigation, Methodology, Validation | Huiping Xu, Investigation, Methodology, Validation | Panyang Wu, Investigation, Methodology, Validation, Formal analysis, Software | Zhiming Xu, Data curation, Investigation, Validation | Xianwei Wang, Conceptualization, Investigation, Methodology, Writing – review and editing, Project administration, Supervision, Validation | Guofu Pi, Conceptualization, Funding acquisition, Validation, Writing – review and editing, Supervision

## DATA AVAILABILITY

Data are available on reasonable request from the corresponding author. The data of gut microbiota by shotgun metagenomic sequencing have been deposited to the SRA database in NCBI (PRJNA1215817). The data of serum metabolites by LC-MS sequencing have been deposited in Metabolights database (MTBLS11685).

## ETHICS APPROVAL

All study procedures were approved by institutional review boards of the First Affiliated Hospital of Zhengzhou University (2022-KY-1476).

## ADDITIONAL FILES

The following material is available online.

## Open Peer Review

**PEER REVIEW HISTORY (review-history.pdf).** An accounting of the reviewer comments and feedback.

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
