## [Reviewer comments · Microbiology Spectrum]

Microbiology Spectrum

***Prevotella copri* mediated caffeine metabolism involves ferroptosis of osteoblasts in osteoarthritis**

Feng Li, Xin Wen, Pu Xue, Huiping Xu, Panyang Wu, Zhiming Xu, Xianwei Wang, and Guofu Pi

Corresponding Author(s): Guofu Pi, The First Affiliated Hospital of Zhengzhou University

Review Timeline:

Submission Date:	June 27, 2024
Editorial Decision:	September 6, 2024
Revision Received:	December 9, 2024
Accepted:	December 18, 2024

Editor: Erik Hom

Reviewer(s): Disclosure of reviewer identity is with reference to reviewer comments included in decision letter(s). The following individuals involved in review of your submission have agreed to reveal their identity: Jiacan Su (Reviewer #1)

Transaction Report:

DOI: <https://doi.org/10.1128/spectrum.01575-24>

Re: Spectrum01575-24 (*Prevotella copri* mediated caffeine metabolism involves ferroptosis of osteoblasts in osteoarthritis)

Dear Prof. Guofu Pi:

Thank you for the privilege of reviewing your work. Below you will find my comments, instructions from the Spectrum editorial office, and the reviewer comments.

Please make sure to address the reviewers's comments carefully and ensure that your data has been deposited at NCBI with accession numbers provided in the revised manuscript. In your cover letter to me upon revision, please summarize what you have changed in the manuscript.

Revision Guidelines

Sincerely,
Erik Hom
Editor
Microbiology Spectrum

Reviewer #1 (Comments for the Author):

This manuscript focused on whether gut microbiota is involved in coffee consumption and OA, and the authors screened out that *P. copri*, an endogenous bacteria, has the ability to metabolize caffeine and revealed the effects of it on OA progression. This

research perspective was relatively innovative, and this manuscript provided a certain degree of knowledge supplementation for related research fields, but it was still necessary to point out several shortcomings.

- 1) The amount of data in this study is relatively abundant, but lacks good arrangement and layout management. It is recommended that the authors reorganize and plan the content of each image and table.
- 2) More background information about the correlation between gut microbiota and OA, as well as age-related degenerative diseases of bone, needs to be mentioned in the Introduction section.
- 3) The composition and layout of the images in the entire manuscript are relatively rough. Please further revise it to meet the publishing standards.
- 4) The description of animal model construction and gut microbiota detection needs to be more specific, especially in addressing the differences between different groups.
- 5) Strengthen the conclusion of this research by summarizing the key findings and its implications more robustly. Discuss potential future research directions that could stem from your findings, particularly in clinical applications or therapeutic development.
- 6) The research significance and potential promotional performance of this study need to be further mentioned and optimized.
- 7) Most recent and related studies such as DOI: 10.1016/j.arr.2024.102196; DOI: 10.1080/19490976.2023.2295432; DOI: 10.1016/j.jot.2022.08.003, etc., are recommended to be cited in proper places.

Reviewer #2 (Public repository details (Required)):

In this research, data generated from fecal metagenome sequencing or serum metabonomics should be deposited in public repository.

Reviewer #2 (Comments for the Author):

The manuscript explores a novel link between gut microbiota (particularly *Prevotella copri*) and caffeine metabolism, and their roles in osteoarthritis (OA) progression. The focus on microbial-mediated caffeine degradation and ferroptosis is original and highly relevant. I have some concerns need to be addressed:

Major Comments: In Figure 1A (PCoA of gut microbiota) and Figure 2A (PLS-DA of serum metabolomics), there is noticeable dispersion within the experimental groups. This variation could indicate underlying heterogeneity in the samples, potentially due to unaccounted variables. The manuscript does not address this issue explicitly, and it would be valuable for the authors to discuss this observation. Given that both gut microbiota composition and serum metabolomic profiles are known to be influenced by a range of factors, such as age, gender, diet, lifestyle, and genetic background, it is crucial to consider whether these factors could contribute to the dispersion seen in the PCoA and PLS-DA models.

Minor Comments: In the Methods, the description in sections including Fecal metagenome sequencing, Serum metabonomics, Transmission Electron Microscopy (TEM) staining, and Statistical analysis are too concise. It should be described in more detail.

Dear Reviewers,

Thank you for your comments concerning our manuscript entitled “*Prevotella copri* mediated caffeine metabolism involves ferroptosis of osteoblasts in osteoarthritis” (ID: Spectrum01575-24). Those comments were all valuable and very helpful for revising and improving our paper, as well as guiding significance to our study. We have carefully studied the comments and have made corrections, which we hope meet with your approval. The revised portions are marked in yellow in the manuscript.

Our point-by-point-responses to the reviewer’s comments are as follows:

Reviewer #1 (Comments for the Author):

1) The amount of data in this study is relatively abundant, but lacks good arrangement and layout management. It is recommended that the authors reorganize and plan the content of each image and table.

Response: We have reorganized and plan the content of the images and tables, particularly Figures 1 and 2.

2) More background information about the correlation between gut microbiota and OA, as well as age-related degenerative diseases of bone, needs to be mentioned in the Introduction section.

Response: The above proposals have been added in the introduction section.

3) The composition and layout of the images in the entire manuscript are relatively rough. Please further revise it to meet the publishing standards.

Response: We have reorganized and plan the rough images in order to meet the publishing standards.

4) The description of animal model construction and gut microbiota detection needs to be more specific, especially in addressing the differences between different groups.

Response: We have rewritten the section of animal model construction and described the method of gut microbiota detection more specific.

5) Strengthen the conclusion of this research by summarizing the key findings and its implications more robustly. Discuss potential future research directions that could stem from your findings, particularly in clinical applications or therapeutic development.

Response: We have rewritten the conclusion by summarizing the key findings and its implications robustly. We have discussed the potential future research directions in clinical applications and therapeutic development for OA patients.

6) The research significance and potential promotional performance of this study need to be further mentioned and optimized.

Response: We have refined the research significance and application prospects in conclusion section.

7) Most recent and related studies such as DOI: 10.1016/j.arr.2024.102196; DOI: 10.1080/19490976.2023.2295432; DOI: 10.1016/j.jot.2022.08.003, etc., are recommended to be cited in proper places.

Response: The above three articles are helpful and cited in the introduction section.

Reviewer #2 (Public repository details (Required)):

In this research, data generated from fecal metagenome sequencing or serum metabonomics should be deposited in public repository.

Response: The data of gut microbiota by shotgun metagenomic sequencing have been

deposited to the SRA database in NCBI (SUB14866088). The data of serum metabolites by LC-MS sequencing have been deposited in Metabolights database (MTBLS11685).

Reviewer #2 (Comments for the Author):

The manuscript explores a novel link between gut microbiota (particularly *Prevotella copri*) and caffeine metabolism, and their roles in osteoarthritis (OA) progression. The focus on microbial-mediated caffeine degradation and ferroptosis is original and highly relevant. I have some concerns need to be addressed:

Major Comments: In Figure 1A (PCoA of gut microbiota) and Figure 2A (PLS-DA of serum metabolomics), there is noticeable dispersion within the experimental groups. This variation could indicate underlying heterogeneity in the samples, potentially due to unaccounted variables. The manuscript does not address this issue explicitly, and it would be valuable for the authors to discuss this observation. Given that both gut microbiota composition and serum metabolomic profiles are known to be influenced by a range of factors, such as age, gender, diet, lifestyle, and genetic background, it is crucial to consider whether these factors could contribute to the dispersion seen in the PCoA and PLS-DA models.

Response: We added the required discussion in the in the conclusion section as “However, there is noticeable dispersion on the characteristics of gut microbiota and serum metabolites within the experimental groups. Further large-scale populationbased cohorts are still needed to comprehensively assess the effects of demographic characteristics, dietary patterns, lifestyle, genetic background, and other factors on gut microbiota and seum metabolites in individuals with coffee consumers.”

Minor Comments: In the Methods, the description in sections including Fecal metagenome sequencing, Serum metabonomics, Transmission Electron Microscopy

(TEM) staining, and Statistical analysis are too concise. It should be described in more detail.

Response: We have described the above four sections in detail.

We sincerely thank the Reviewers for these good comments.

We have tried our best to improve the manuscript and made some changes in the manuscript. These changes will not influence the content and framework of the paper.

We appreciate for Reviewers' warm work earnestly, and hope that the correction will be met with approval. If further modification is required, we will work hard to complete it as soon as possible.

Once again, thank you very much for your comments and suggestion.

Best regards,

Guofu Pi, and Xianwei Wang

Department of Orthopedics, the First Affiliated Hospital of Zhengzhou University,
Zhengzhou 450000, China

Re: Spectrum01575-24R1 (*Prevotella copri* mediated caffeine metabolism involves ferroptosis of osteoblasts in osteoarthritis)

Dear Prof. Guofu Pi:

Your manuscript has been accepted, and I am forwarding it to the ASM production staff for publication. Your paper will first be checked to make sure all elements meet the technical requirements. ASM staff will contact you if anything needs to be revised before copyediting and production can begin. Otherwise, you will be notified when your proofs are ready to be viewed.

Sincerely,
Erik Hom
Editor
Microbiology Spectrum